# A novel lineage-tracing mouse model for studying early MmuPV1 infections

**Vural Yilmaz[1], Panayiota Louca[1], Louiza Potamiti[2], Mihalis Panayiotidis[2], Katerina Strati[1]\***

[1]Department of Biological Sciences, University of Cyprus, Nicosia, Cyprus; [2]Department of Cancer Genetics, Therapeutics & Ultrastructural Pathology, The Cyprus Institute of Neurology & Genetics, Nicosia, Cyprus

**Abstract** Human papillomaviruses are DNA viruses that ubiquitously infect humans and have been associated with hyperproliferative lesions. The recently discovered mouse specific papillomavirus (MmuPV1) provides the opportunity to study papillomavirus infections in vivo in the context of a common laboratory mouse model (*Mus musculus*). To date, a major challenge in the field has been the lack of tools to identify, observe, and characterize individually the papillomavirus hosting cells and also trace the progeny of these cells over time. Here, we present the successful generation of an in vivo lineage-tracing model of MmuPV1-harboring cells and their progeny by means of genetic reporter activation. Following the validation of the system both in vitro and in vivo, we used it to provide a proof-of-concept of its utility. Using flow-cytometry analysis, we observed increased proliferation dynamics and decreased MHC-I cell surface expression in MmuPV1-treated tissues which could have implications in tissue regenerative capacity and ability to clear the virus. This model is a novel tool to study the biology of the MmuPV1 host-pathogen interactions.

## Editor's evaluation

This paper describes the development of an interesting and novel model permitting lineage tracing using the MmuPV1 mouse papillomavirus. This model will allow analysis of the temporal and spatial dynamics of MmuPV1 infection replication and assembly in different tissue sites and is a valuable contribution to the understanding of papillomavirus biology.

**\*For correspondence:** strati@ucy.ac.cy

**Competing interest:** The authors declare that no competing interests exist.

## Introduction

Human papillomaviruses (HPVs) are DNA viruses which are linked to 5% of human cancers (***McBride, 2017***) and are also responsible for commensal infections. Mucosotropic high-risk HPVs, a group which includes HPV16 and HPV18, have been associated with malignancies, mainly cervical cancer, other anogenital cancers, and a subset of head and neck squamous cell carcinoma (***zur Hausen, 2009***). For this reason, the contribution of these viruses to cancer has been extensively studied with significant advances in cancer prevention, prophylaxis, and detection. In addition to representing a significant source of pathogenesis, it is clear that papillomaviruses also abundantly colonize human epithelia without causing an apparent pathology (***Hannigan et al., 2015***), an aspect of their biology which is more poorly understood.

Since differentiation of stratified epithelia is essential for the productive HPV replication cycle (***Roden and Stern, 2018***), and due to the species specificity of papillomaviruses, it has been challenging to study the host-virus interaction and the progress of infection of HPV in the laboratory. Furthermore, the outcome of HPV infection is greatly influenced by the host immune system and inflammation induced by tissue damage that is required in order to expose the underlying basal

layer to the virus (*Amador-Molina et al., 2013*). Thus, the best system to study virus-host interaction taking into consideration all the factors that influence infection outcome is to explore this in vivo.

This is now possible due to the discovery of a mouse specific papillomavirus (MmuPV1) (*Ingle et al., 2011*; *Hu et al., 2017*). This mouse papillomavirus provides the opportunity to study papillomavirus infections in the context of a small common laboratory animal for which abundant reagents are available and for which many strains exist. Even though the genome of the human and mouse papillomavirus is similar, with the exception of the lack of E5 gene in the mouse virus, HPVs, and MmuPV1 do indeed show differences at the level of nucleotide sequences (*Joh et al., 2011*). Nevertheless, by studying the mouse papillomavirus-host interaction, we can gain a better understanding on how the presence of the papillomavirus alters the host cell behavior over time (*Spurgeon and Lambert, 2020*).

Existing approaches in the field have relied on long-term phenotypes leading to clearer understanding regarding the mechanisms of oncogenesis. This remains true in the development of a novel mouse model of spontaneous high-risk HPV E6/E7-expressing carcinoma (*Henkle et al., 2021*), an inducible mouse model of HPV oncogene expression (*Böttinger et al., 2020*) and other novel models based on infection (*Brendle et al., 2021*; *Spurgeon and Lambert, 2020*; *Wei et al., 2020*) and transmission (*Spurgeon and Lambert, 2019*). Reliance on cytopathic effects has resulted in a severe understudy of the majority of the papillomavirus infections which do not lead to overt signs of infection. Additionally, subclinical MmuPV1 infections have also been previously reported (*Hu et al., 2017*; *Xue et al., 2017*). These infections remain critical to study as they can offer significant insights on the prevention of substantial morbidities such as laryngeal respiratory papillomatosis, genital wart recurrence etc. (*Egawa and Doorbar, 2017*). This underscores the clear need for a model where one can observe directly the MmuPV1 infected cells and furthermore, trace them over time during the course of the infection.

Here, we have designed and developed a model for in vivo lineage tracing of the cells initially harboring MmuPV1. We have created an MmuPV1-lox-Cre-lox plasmid which we deliver to the tail skin of reporter R26R-lox-STOP-lox-eYFP mice. Expression of self-deleting Cre from the plasmid results in recombination of the loxP sites and excision of the sequence flanked by the loxP sites. Therefore, introduction of the plasmid in mouse cells leads to the production of Cre recombinase and loss of the Cre sequence and its promoter from the plasmid and, more importantly, recircularization of the MmuPV1 genome, and the loss of the stop codon upstream of the yellow florescence protein (YFP) gene. This lets us observe and trace longitudinally, cells that initially harbored the plasmid containing the viral genome. Since YFP expression is genetically activated, it is conserved in the following progeny cells of the initial cell that taken up the plasmid. This model provides, for the first time, a promising tool to study the biology of MmuPV1 infection and its target cells along with the impact of the virus on these cells.

## Results and discussion
### Cre expression from MmuPV1-lox-Cre-lox leads to reporter activation and plasmid recombination in vitro

To validate the generated MmuPV1 plasmid (*Figure 1E*) in vitro, we transfected mouse embryonic fibroblasts (MEFs) (isolated from R26R-lox-STOP-lox-eYFP mice) with the generated MmuPV1-lox-Cre-lox plasmid (MmuPV1-Cre) or with the control plasmid, pBABE-GFP. As expected, we detected the function of the Cre recombinase which facilitates the YFP expression in the MmuPV1-lox-Cre-lox transfected cells by immunofluorescence (IF) (*Figure 1A*). Additionally, we isolated DNA from these transfected cells and determined the PCR amplification of the Cre sequence and sequence flanking the Cre cassette (Cre loss), showing the recombination of the MmuPV1-lox-Cre-lox plasmid after Cre expression (*Figure 1B*).

Lastly, we detected the presence of MmuPV1 viral transcripts E1/E4, L1 and E6 via RT-PCR analysis in MmuPV1-lox-Cre-lox transfected cells but not in pBABE-GFP transfected cells (*Figure 1C–D*). Together, these results validate that the constructed MmuPV1-lox-Cre-lox plasmid can indeed lead to reporter activation, Cre-excision, and MmuPV1 genome recircularization in vitro.

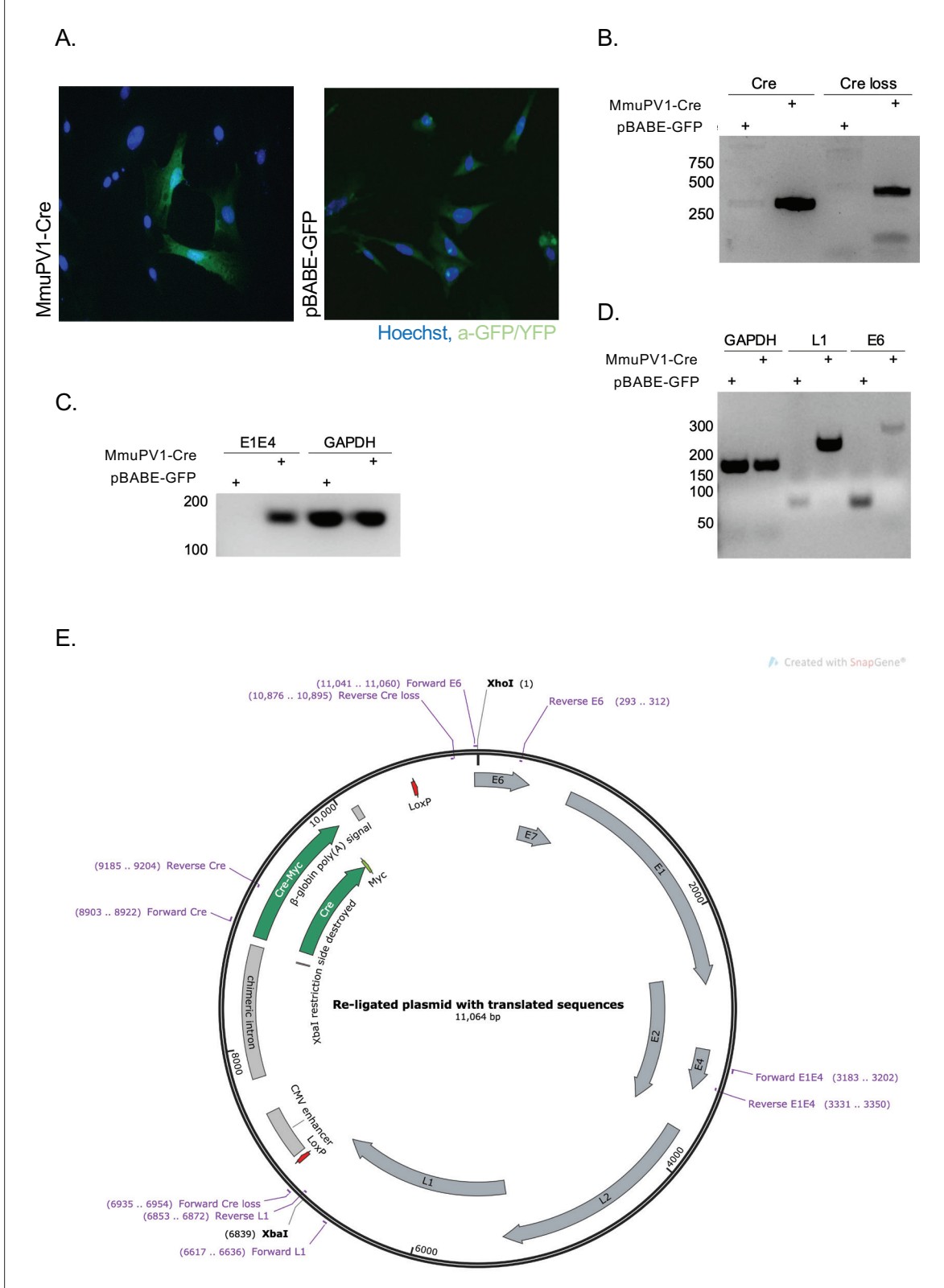

**Figure 1.** Cre expression from MmuPV1-lox-Cre-lox leads to plasmid recombination in vitro. (**A**) Immunofluorescence of mouse embryonic fibroblasts (MEFs) (isolated from R26R-EYFP mice) transfected with pBABE-GFP or MmuPV1-lox-Cre-lox (MmuPV1-Cre) plasmids 48 hr post-transfection (green: a-GFP/YFP, blue: Hoechst). (**B**) DNA isolated from transfected cells was used for PCR amplification of the Cre sequence and sequence flanking the Cre cassette to determine recombination of the MmuPV1-lox-Cre-lox plasmid after Cre expression. (**C–D**) RT-PCR analysis for the presence of the viral

*Figure 1 continued on next page*

Figure 1 continued

transcripts E1E4 (C) L1 and E6 (D) in GFP and MmuPV1-lox-Cre-lox transfected MEFs. (**E**) Map of the MmuPV1-lox-Cre-lox plasmid after re-ligation and translated sequences. Created with SnapGene software (from Insightful Science; available at snapgene.com).

## Cells which take up MmuPV1-lox-Cre-lox plasmid can be detected and traced over time in vivo

Next, we used the plasmid in mice. It is known that infection with MmuPV1 is mainly cleared by the functions of the CD8+T cell-dependent adaptive immunity (*Handisurya et al., 2014*). Therefore, immunocompromised mice are more susceptible to the mouse papillomavirus. Thus, we transiently immunocompromised mice before delivering the MmuPV1-lox-Cre-lox plasmid by using UV-B radiation. UV-B was shown to suppress the adaptive immune response, whereas the innate immune response remains largely unaffected (*Uberoi et al., 2016*).

Following the UV-B radiation, the base of the tail skin of R26R-lox-STOP-lox-eYFP mice was superficially abraded with a pipette tip to expose the tail basal layer. Then, MmuPV1-lox-Cre-lox plasmid was delivered and skin samples were harvested at the specified time points to check the presence of YFP expression by IF (*Figure 2A*).

We observed characteristic clones of YFP-expressing cell population in the skin indicative of heritable YFP activation in a skin basal cell. Clones were most prominent at day 3 and day 5 after plasmid delivery (*Figure 2B*, indicated with white lines). Additionally, we were able to follow-up YFP+ cells at later time points. These results indicate that our model can indeed be used as an in vivo lineage-tracing model for MmuPV1, where virus-hosting cells and their progeny can be individually observed via their genetically induced YFP expression.

As a control for our lineage-tracing model, we use CAG-Cre plasmid which drives the expression of a self-deleting Cre recombinase, just as in the MmuPV1-lox-Cre-lox plasmid, but it does not encode any MmuPV1 genes. As performed above, pCAG-Cre was delivered to R26R-lox-STOP-lox-eYFP mice after UV-B irradiation and tail skin scarification. Tail skins were isolated and treated in an identical manner as MmuPV1-treated tissues (*Figure 2C*, indicated with white lines).

## Tissues treated with MmuPV1-lox-Cre-lox plasmid show evidence of active MmuPV1 gene transcription in vivo

We also examined the presence of the viral genes and transcripts in the cells of the same tissue samples and same time points as above. Firstly, we used the RNA in situ hybridization assay of RNAscope to check for the expression of MmuPV1 transcripts, E1/E4 and E6/E7. We observed that in the cells that have taken up the MmuPV1-lox-Cre-lox plasmid, E6/E7 transcripts started to appear as early as 3 days post-delivery (p.d.) and peaked around day 7 p.d. (*Figure 3A*), whereas the E1/E4 transcripts were detected at day 5 p.d. onward and peaked at day 10 p.d. (*Figure 3B*). Furthermore, we did not detect any of the tested MmuPV1 viral transcripts in the cells that have taken up the CAG-Cre plasmid since the viral genome of MmuPV1 is lacking in this control plasmid (*Figure 3C–D*). Additionally, we distinguished the viral RNA signal from the viral genomic DNA signal by treating the tissue sections with DNAse and/or RNAse prior to probe hybridization in which we confirmed that the majority of the observed signal for MmuPV1-E6/7 and E1/E4 was indeed viral RNA signal (*Figure 3—figure supplement 1A, B*).

To confirm the presence of genome on the skin surface, we swabbed the surface of the treated tail skins and washed the swabs to remove cells prior to collecting the skin tissue for the above-mentioned assays. To verify the anticipated recircularization of the MmuPV1 plasmid occurred in vivo as well, we isolated DNA to determine the PCR amplification of the sequence flanking the Cre cassette (Cre loss) (*Figure 3E*, *Figure 3—figure supplement 1C*). We also assessed the presence of the L1 gene of MmuPV1 via PCR as a way of confirming the presence of the viral genome on the skin surface; GAPDH was used as a housekeeping gene (*Figure 3E*, *Figure 3—figure supplement 1C*).

As expected, at day 0, we did not observe any viral transcripts, genes, or the Cre loss sequence. However, we detected the Cre loss and the MmuPV1-L1 sequences at day 5 and day 10 samples of the skin surface (*Figure 3E*), whereas CAG-Cre control samples showed no signal as expected (*Figure 3—figure supplement 1C*). The presence of the recircularized MmuPV1 plasmid on the surface of the skin could be a result of the eventual migration of the progeny cells to the superficial layer from the

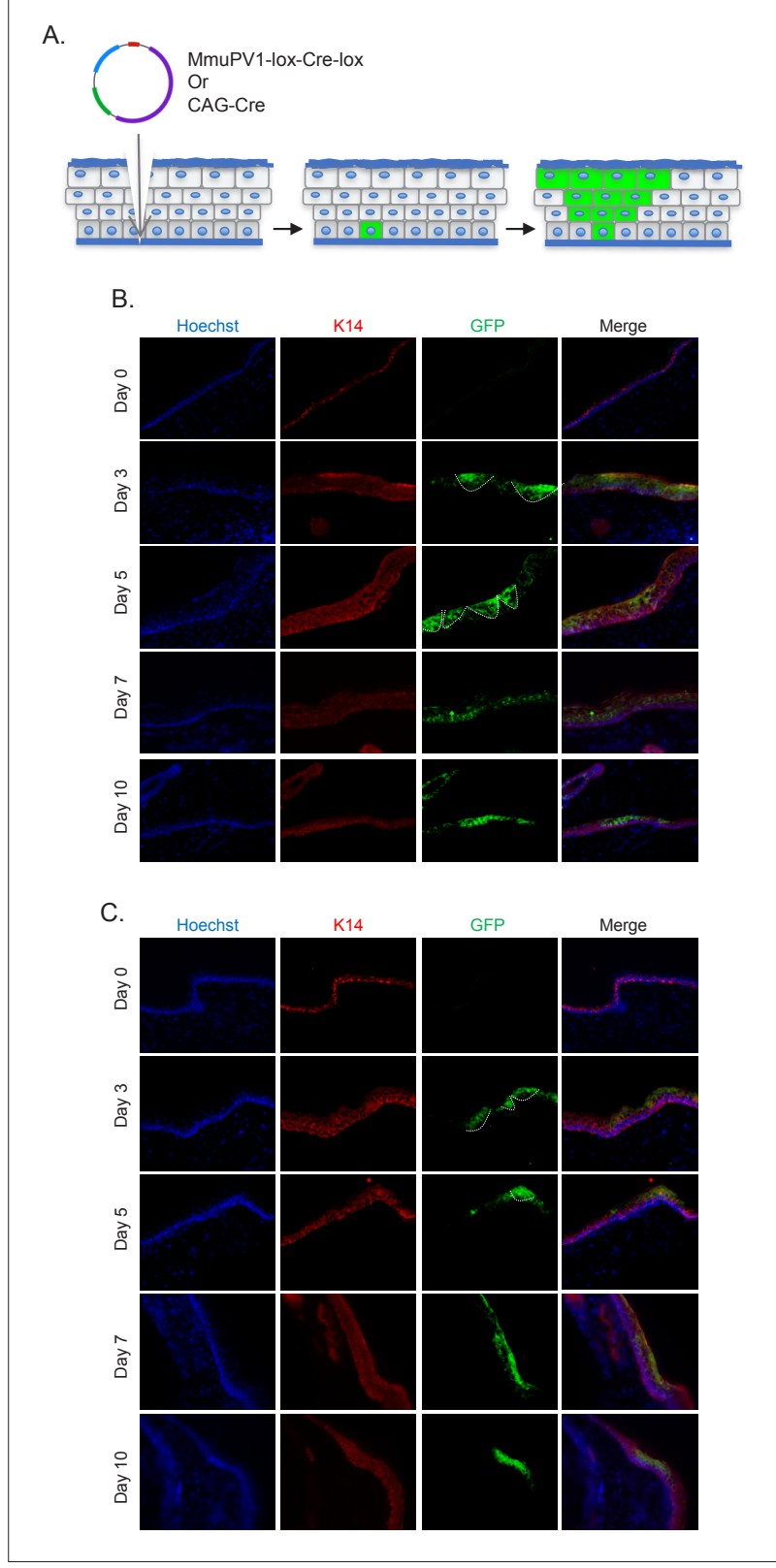

**Figure 2.** Cells which taken up MmuPV1-lox-Cre-lox plasmid can be detected and traced over time in vivo. (**A**) Experimental design; the MmuPV1-lox-Cre-lox plasmid is delivered to the tail skin of R26R-lox-STOP-lox-eYFP mice (of pure C57BL/6 genetic background) after a UV-B irradiation and a superficial scarification. The expected observation is an almost 'V' shaped YFP expressing cell population where a single or a couple of basal cells have

*Figure 2 continued on next page*

*Figure 2 continued*

taken up the plasmid at the bottom and went through cell division where progenies move upward in the skin layers toward the surface. (**B**) Immunofluorescence (IF) of MmuPV1-lox-Cre-lox plasmid delivered tail skin section at the indicated time points (red: K14, green: a-GFP/YFP). (**C**) IF of CAG-Cre plasmid delivered tail skin section at the indicated time points (red: K14, green: a-GFP/YFP). White lines indicate the approximate 'V' shaped YFP expressing cell populations on day 3 and day 5 only.

initially MmuPV1-harboring cell at the basal layer and shedding the viral particles on the skin surface. However, since the level of the detected signal is considerably low, as can be seen in *Figure 3E* or *Figure 3—figure supplement 1C*, we believe that the amount of correctly recombined MmuPV1-lox-Cre-lox plasmid after Cre expression is rather low but still present in vivo. At day 30 time point, we did not detect any viral genome on swabs of the tail skin even though we still detect YFP+ cells at this time point, albeit at low levels (*Figure 4A*). This might be indicative of the greatly reduced presence (below limit of detection) or clearance of the virus on the skin.

Furthermore, we checked whether the formation of virions is occurring in our system following the initial plasmid delivery and the consequent recombination and genome recircularization. Using transmission electron microscopy (TEM), we were able to visualize virion-like structures in the perinuclear area of epithelial cells in all the MmuPV1-Cre treated mice, whereas none of these structures were observed in mice treated with the control CAG-Cre plasmid (*Figure 3F*). Together, we were able to detect the MmuPV1 transcripts of E1/E4 and E6/E7 in the cells which harbor the MmuPV1-lox-Cre-lox plasmid and also confirmed the recombination of the MmuPV1-lox-Cre-lox plasmid after Cre expression by detecting the sequence flanking the Cre cassette (Cre loss).

## Progeny of cells which received MmuPV1-lox-Cre-lox plasmid exhibit higher proliferation dynamics and lower surface expression of MHC-I in vivo

To provide proof of principle that our model system may be used to answer question regarding papillomavirus biology and taking advantage of the ability to detect YFP expressing cells, we applied the plasmids to the tail skin of R26R-lox-STOP-lox-eYFP mice as mentioned above. Then, skin samples were harvested at the specified time points and processed and stained for flow cytometric analysis.

Firstly, we applied a standard gating strategy to exclude dead and doublet cell populations (*Figure 4—figure supplement 1A*). Then, we detected the YFP expressing cells and compared numbers and frequencies between groups and different time points (*Figure 4A*, *Figure 4—figure supplement 1B*). We found that both numbers and percentages of YFP+ cells were much higher in MmuPV1 samples at day 5 p.d., which decreased overtime as seen at day 10 and day 30 p.d. Whereas, pCAG-Cre samples only start to have increased YFP+ cells around day 10 p.d. (*Figure 4A*, *Figure 4—figure supplement 1B*). These results are also in line with the IF data we saw previously and can indicate to a difference in the actual amount of plasmid delivered to the tissue or the different dynamics of the plasmids due to the presence of the MmuPV1 genes. Furthermore, coinciding with the increased number of YFP+ cells, we observed that MmuPV1-harboring cells exhibit significantly increased levels of Ki-67 around day 5 p.d. compare to other time points (*Figure 4B*, *Figure 4—figure supplement 1C*). We further directly visualized Ki-67-positive cells by immunohistochemistry (IHC) in the regions of infected skin where we previously detected the MmuPV1 transcripts via RNAscope and observed that the cells harboring MmuPV1-Cre plasmid indeed exhibit higher proliferation around day 5 p.d. compare to the cells harboring CAG-Cre control plasmid (*Figure 4D*).

Papillomaviruses have previously been reported to reduce MHC-I expression, but evidence was generated in vitro (*Georgopoulos et al., 2000*; *Heller et al., 2011*; *Li et al., 2010*). To evaluate the levels of MHC-I specifically on the cell surface of the MmuPV1-harboring cells, we used flow-cytometry to interrogate MHC-I levels within the YFP+ population. Interestingly, we observed significantly lower expression of MHC-I molecules on the YFP+ cells of MmuPV1-treated tissues at day 5 p.d., whereas YFP+ cells in control treated tissues exhibit increased MHC-I expression at this time point, presumable due to YFP neo-antigen expression (*Figure 4C*, *Figure 4—figure supplement 1D*). These effects waned at later time points. Our results agree with previous in vitro studies and indicate a possible immune evasion strategy by the MmuPV1 where the virus is causing a decreased expression of MHC-I molecules on the cell surface to reduce the antigen exposure to the immune cells.

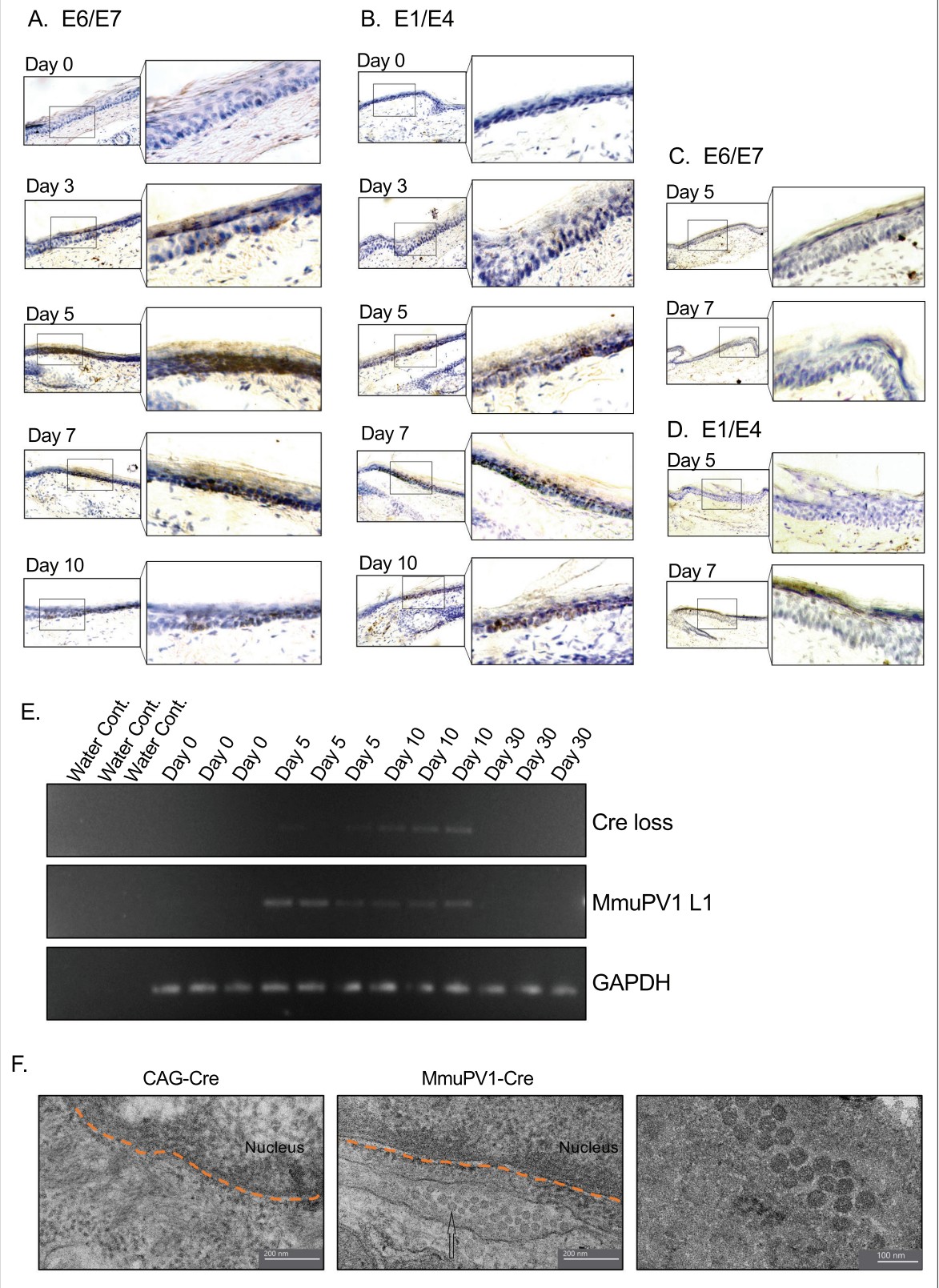

**Figure 3.** Tissues with MmuPV1-lox-Cre-lox plasmid express viral transcripts in vivo. (**A–B**) RNAscope of the MmuPV1-lox-Cre-lox plasmid delivered tail sections of the R26R-lox-STOP-lox-eYFP mice at the indicated time points to assess the presence of viral transcripts E6/E7 and E1/E4, respectively. (**C–D**) RNAscope of the CAG-Cre plasmid delivered tail sections of the R26R-lox-STOP-lox-eYFP mice at the indicated time points to assess the presence of viral transcripts E6/E7 and E1/E4, respectively. The presence of the transcripts appears as brown dots. (**E**) DNA isolated from the swap samples of the

*Figure 3 continued on next page*

*Figure 3 continued*

MmuPV1-lox-Cre-lox plasmid delivered tail sections of the R26R-lox-STOP-lox-eYFP mice and was used for PCR amplification of the sequence flanking the Cre cassette (Cre loss) and L1 of MmuPV1 to determine the presence and the recombination of the MmuPV1_Cre plasmid after Cre expression. (**F**) Representative electron micrographs showing the control (CAG-Cre) (left panel) and infected (MmuPV1-Cre) (middle and right panel) tail tissue samples, respectively. Arrow shows the formed virion-like structures in the MmuPV1-Cre samples which are absent in all the control samples. The dotted orange line highlights the border of the nucleus. Thin sections (0.1 μm) were examined under a transmission electron microscopy (TEM) at 120 kV (TALOS L120C).

The online version of this article includes the following figure supplement(s) for figure 3:

**Figure supplement 1.** RNAscope controls with DNAse and/or RNAse and CAG-Cre controls for the detection of MmuPV1-L1 and Cre loss sequences.

It is important to note, that the differences in proliferation and MHC- I expression were specific to the YFP + cells within the tissue. Also, within the same tissues, we analyzed the Ki-67 and MHC-I levels of the neighboring YFP- cells which have not taken up the given plasmid (*Figure 4—figure supplement 2*). We observed no significant difference between time points or between the YFP- cell population of the MmuPV1 plasmid-treated tissue and the CAG-Cre plasmid-treated tissue (*Figure 4—figure supplement 2A, B*). These results suggest that any differences observed are specific to plasmid uptake.

Together, these results indicate that during early acute infection, MmuPV1 causes increased proliferation which may be accompanied by decreased antigen presentation. Additionally, here we show that our novel in vivo lineage-tracing model can be used as a tool to answer such key questions in the field of MmuPV1 biology.

## Conclusions and significance

Here, we describe a novel in vivo lineage-tracing model for MmuPV1 infected cells in laboratory mice (*Mus musculus*). We propose that the ability to trace MmuPV-1 harboring cells and their progeny by microscopy, as well as to quantify and analyze by means of flow-cytometry renders this model a suitable and convenient tool for answering a variety of outstanding questions in papillomavirus biology (*Lambert et al., 2020*). Particularly, we believe that the novel model described here can be used to investigate the early effects of papillomavirus on the target cells, transcriptional and phenotypic changes of the papillomavirus-harboring cells during the acute phase of the infection, and investigate the initial viral-host immune interactions.

## Materials and methods
### Plasmids

To generate a plasmid with a self-deleting Cre sequence and the full MmuPV1 genome, we combined sequences from four plasmids: puC19-MmuPV1 (gift from Paul F. Lambert), pCAG-cre (a gift from Connie Cepko (Addgene plasmid # 13775; http://n2t.net/addgene:13775; RRID: Addgene_13775)), small_pMA-T, and large_pMA-T (synthesized by Invitrogen). To construct the final plasmid, we first PCR amplified the sequence in large_pMA-T that contains the lox and part of the E6 sequence using primers to introduce PstI and HindIII restriction sites in the 5' and 3' ends of the sequence, respectively. Using PstI and HindIII, we cloned lox-E6 in the pCAG-Cre plasmid. In this new plasmid, we subcloned the L1-lox sequence from small_ pMA-T plasmid using blunt end ligation (L1-Lox sequence was isolated using SacI and KpnI restriction enzymes, the Cag-cre plasmid was digested with SalI and T4 DNA polymerase was used to generate blunt ends). We refer to this plasmid as plox-cre-lox which was sequenced to confirm the correct insertion of L1-lox-Cre-lox-E6 sequence. To generate the plasmid for animal infection, we removed the ampicillin bacterial selection cassette from lox-cre-lox and MmuPV1 plasmids using XhoI and XbaI. The two fragments (lox-cre-lox and MmuPV1) were gel purified and ligated overnight at 16 °C (ligation was done using a ratio of 1:2.5, MmuPV1 to lox-cre-lox). Complete plasmid sequences are provided in the supplementary file. pBABE GFP (a gift from William Hahn [Addgene plasmid # 10668; http://n2t.net/addgene:10668; RRID:Addgene 10668]) plasmid was used as a control for the in vitro experiments.

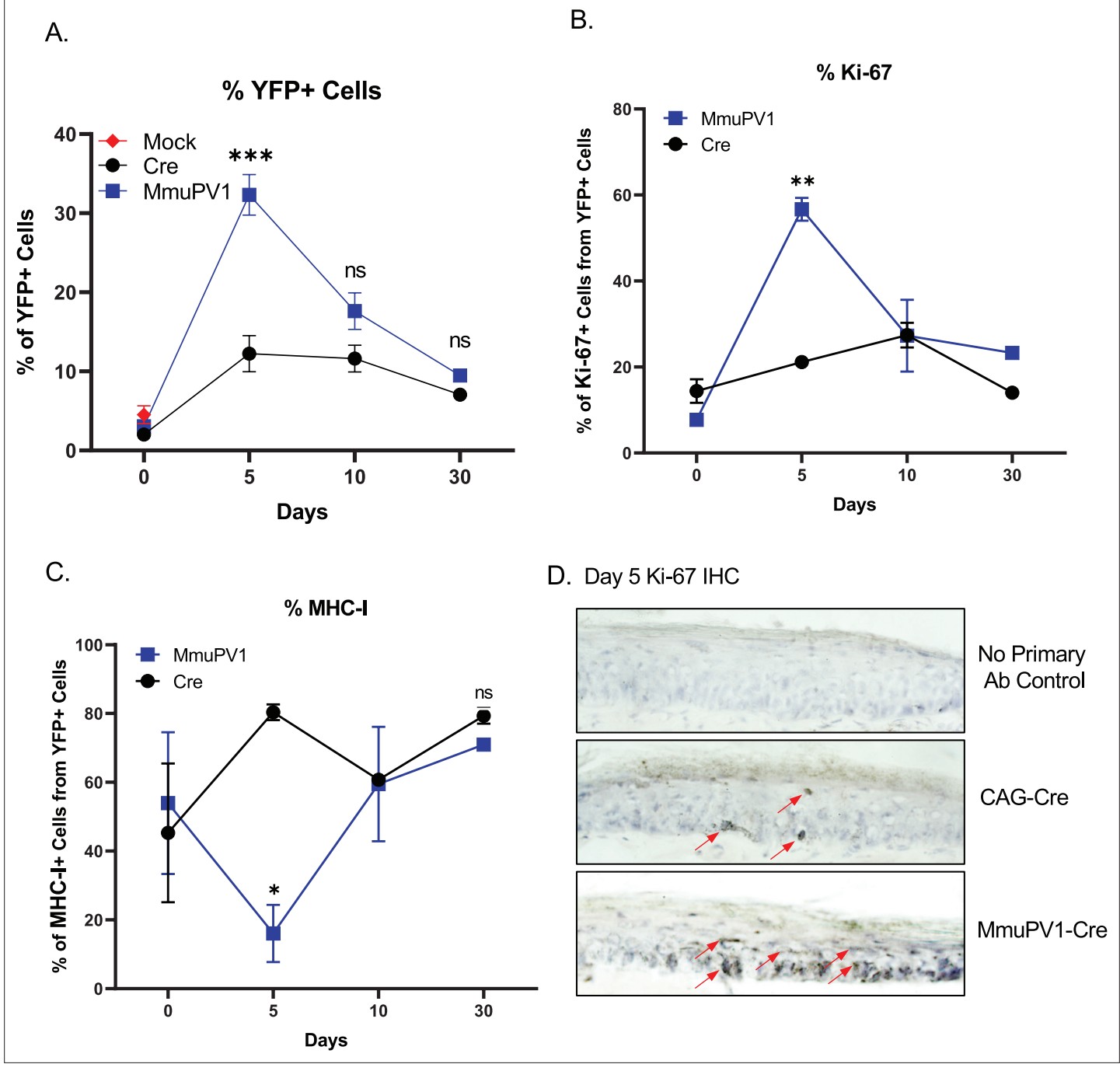

**Figure 4.** Cells with MmuPV1-lox-Cre-lox plasmid have increased proliferation rate and decreased MHC-I expression on the cell surface. (**A**) Percentages of YFP+ cells. (**B**) Percentages of Ki-67+ cells out of only YFP+ cells. (**C**) Percentages of MHC-I+ cells out of only YFP+ cells. Mock infection with PBS (red square) was only performed for day 0 time point. (**D**) Representative immunohistochemistry images for Ki-67 staining (indicated by red arrows) on tail tissue sections at day 5 of post-delivery of CAG-Cre control or MmuPV1-Cre plasmid. Data representative of two independent experiments with n = 3 mice per group. The results are presented as mean ± SEM. Not significant (ns), * p < 0.05, ** p < 0.01 and *** p < 0.001 by nonparametric 2-way ANOVA test.

The online version of this article includes the following source data and figure supplement(s) for figure 4:

**Source data 1.** The source data of FACS for *Figure 4* and its supplements is included in a Microsoft Excel spreadsheet called "Source data - FACS".

**Figure supplement 1.** Cells with either MmuPV1-Cre or CAG-Cre control plasmid can be detected and analyzed by flow cytometry.

**Figure supplement 2.** YFP- cells in the tissues treated with MmuPV1-Cre and CAG-Cre control plasmid have similar Ki-67 and MHC-I expression.

## Tissue culture, transfection, PCR, and RT-PCR

MEFs were used for in vitro transfections. MEFs were isolated from C57BL/6 R26R-lox-STOP-lox-eYFP pregnant mice at 13.5 days post-conception. The embryos were harvested from the mouse uterus and placed washed in PBS (GibcoTM) and penicillin-streptomycin (P/S) (Gibco) antibiotics. The embryos were transferred in a 10 cm dish containing PBS and P/S, and all the membranes were removed to allow embryo isolation. The embryo's head was cut below the eye and visible viscera such as the liver were removed. The embryo was then transferred to a fresh 10 cm dish containing 0.05% trypsin-EDTA ×1 (Gibco) in PBS and was chopped with a razor blade. The dish was incubated in a 37 °C incubator for 20 min, and the cells were pipetted up and down until a single-cell suspension was obtained which was also incubated for another 20 min. Media was added in the dish, mixed well, and the cells were then transferred to T75 bottles until confluent. MEFs were maintained in Dulbecco's Modified Eagle Medium (DMEM, Gibco, cat. no 41965039) supplemented with 10% v/v fetal bovine serum (FBS, Gibco, cat. no 10500064) and 1% v/v penicillin/streptomycin (Gibco, cat. no 15070063). In vitro transfections were performed with FuGENE6 Transfection Reagent (Promega, cat. no E2691) as described previously (*Hong et al., 2009*).

DNA and RNA isolations were performed using TRIZOL (Invitrogen) following the manufacturer's instructions. We used RT-PCR to confirm expression of Cre (Cre forward primer: 5′-ACG AGT GAT GAG GTT CGC AA-3′, Cre reverse primer: 5′-CGC CGC ATA ACC AGT GAA AC-3′, GAPDH was used as a housekeeping gene) and PCR to determine lox-recombination and removal of the sequence flanked by loxp sites from the plasmid (Cre loss forward primer: 5′-ACT GTT TAC TGG GGG CTT AC -3′ and cre loss reverse primer: 5′-GGC AGA ACA CAA TGG AAC TC-3′). RT-PCR analysis was performed to assess the presence of viral transcripts from pBABE GFP and MmuPV1-lox-Cre-lox transfected MEFs. The following primer pairs were used and data were analyzed with Rotorgene 6000 Series Software (Qiagen): GAPDH: 5′-ACT CCA CTC ACG GCA AAT TC-3′ and 5′-TCT CCA TGG TGG TGA AGA CA-3′, E1/E4: 5′-GAA CTC TTC CCA CCG ACA CC-3′ and 5′-AAG GTC CTG CAG ATC CCT CA-3′, E6: 5′-ATG GAA ATC GGC AAA GGC TA-3′ and 5′-CTT TTC AGA GGC AGT AAG GA-3′, L1: 5′-TAT ATA ACA TCA TCG GCA AC-3′ and 5′- TGC TTC CCC TCT TCC GTT TT-3′.

## Mice

R26R-lox-STOP-lox-eYFP (Jackson lab stock number #006148, https://www.jax.org/strain/006148) immunocompetent mice, obtained from the mouse facility of the Cyprus Institute of Neurology and Genetics (CING), were of a pure C57BL/6 genetic background. All mice were used at 6–12 weeks of age and were sex- and age-matched within experiments. All the genotypes were confirmed by means of PCR. Mice were housed at the University of Cyprus, in accordance with regulations and protocols approved by the Cyprus Ministry of Agriculture.

## Plasmid delivery

The back skin of 6–9 weeks old female R26R-lox-STOP-lox-eYFP mice were shaved and a single dose of 1000 mJ/cm$^2$ UV-B irradiation was applied for immunosuppression as previously described (*Dorfer et al., 2021*). Following the UV-B radiation, the mice were given at the same day with 2 × 10$^9$ DNA copies of MmuPV1-lox-Cre-lox or CAG-Cre plasmid in a 10 µl of PBS solution applied at the base of the tail skin by a superficial scarification with a pipette tip to expose the tail basal layer. Tissue samples were then harvested at the specified time points. Tail skin was selected for plasmid delivery for practical reasons (ability to track area where plasmid was delivered and fewer hairs which place barriers on cell dissociation for FACS experiments).

## Tissue procurement and processing

Tail tissue sections from the base of the tail were harvested in a rectangular manner and were cut in half, washed with ×1 PBS, fixed in 4% paraformaldehyde (PFA), and embedded in optimal cutting temperature compound and frozen on dry ice before storing in –80°C. Frozen tissues were then sectioned (17 microns thick) using a cryostat and mounted on the positively charged Thermo Scientific SuperFrost Plus Adhesive slides.

## RNA in situ hybridization

MmuPV1 viral transcripts were detected using RNAscope 2.5 HD Assay-Brown (cat. no. 322300) (Advanced Cell Diagnostics, Newark, CA) according to manufacturer instructions with probes specific for MmuPV1 E6/E7 (Cat #409771) and for MmuPV1 E1/E4 (Cat #473281) as described previously (*Spurgeon and Lambert, 2019*; *Xue et al., 2017*). Tissue sections were treated following protease treatment for 30 min at 40°C followed by the probe hybridization. Tissues were then counterstained with hematoxylin and mounted in Cytoseal media (Thermo Fisher Scientific). To distinguish viral RNA signal from viral genomic DNA signal, tissue sections were treated following protease treatment and prior to probe hybridization with 20 units of DNAse I (Thermo Fisher Scientific, #EN0521), or DNAse I combined with 500 µg RNAse A (Qiagen, #1006657) plus 2000 units RNAse T1 (Fermentas, Waltham, MA, #EN0542) for 30 min at 40 °C.

## Immunofluorescence and immunohistochemistry

For IF, after antigen retrieval was performed in a microwave using 10 mM citrate buffer, the tissue sections were permeabilized and blocked at room temperature for 1 hr with a blocking buffer containing 0.5% skim milk powder, 0.25% fish skin gelatin, and 0.5% Triton X-100. Then, sections were washed with PBS and stained with purified primary antibodies in blocking buffer at 4°C overnight. Tissues were then washed with ×3 PBS, stained with secondary antibodies at room temperature for 1 hr, counterstained with Hoechst dye, mounted in Prolong mounting media (Prolong Gold Antifade reagent, Invitrogen, cat. no. P36930), and sealed with nail polish. Similar approach was used for the in vitro experiments where cultured cells were stained with anti-GFP/YFP antibody to enhance the YFP signal and Hoechst dye to label DNA as mentioned above.

For immunohistochemical (IHC) staining, tissue sections were fixed again in 4% PFA at 4°C for 15 min followed by an antigen retrieval step in a microwave using 10 mM citrate buffer. Quenching of endogenous peroxidase activity was accomplished by incubation with 1% hydrogen peroxidase. Sections were blocked with 5% Normal Horse Serum and stained with Ki-67 recombinant rabbit mono-clonal antibody (Thermo Fisher Scientific, MA5-14520, 1:100) in blocking buffer at 4°C overnight. Then, sections were washed in PBS and incubated with horse radish peroxidase (HRP)-conjugated goat-α-rabbit secondary antibody (Thermo Fisher Scientific, 31460, 1:250). Reaction product was visualized with a diaminobenzidine (DAB) peroxidase substrate kit (SK-4100; Vector Laboratories, Burlingame, CA).

The following antibodies were used for detecting mouse antigens by immunofluorescent staining: rabbit-anti-K14 (polyclonal, 1:1000, Covance, PRB-155b), goat-anti-GFP (polyclonal, 1:100, SICGEN, AB0020-200), Alexa Fluor 488 donkey-anti-goat (polyclonal, 1:250, Jackson, 705-545-147), Alexa Fluor 488 donkey-anti-mouse (polyclonal, 1:250, Jackson, 715-545-150), FITC donkey-anti-rabbit (polyclonal, 1:250, Jackson, 711-095-152), and Alexa Fluor 594 donkey-anti-rabbit (polyclonal, 1:250, Jackson, 711-585-152).

## Flow cytometry

After mice were sacrificed and tail skins were extracted as described above, the tail samples were incubated in a 0.25% trypsin containing EDTA solution (Sigma-Aldrich) overnight at 4°C. The next day, epidermis was separated from dermis by using tweezers, the tissue was cut into small pieces and was mechanically dissociated by using a handheld homogenizer (POLYTRON PT 1200 E). Subsequently, the resulting single cell suspensions were washed with PBS and cells were incubated with fluorescently conjugated antibodies for 30 min at 4°C. After marker staining, cells were fixed and permeabilized for 10 min with 2 × BD lysing solution (BD Biosciences) + 0.1% Tween, washed, and then stained in PBS for intracellular Ki-67 staining. The following antibodies (indicating target-fluorochrome) were purchased from Biolegend and were used according to the manufacturer's instructions: MHC-I(H-2Kb)-Brilliant violet 421, Ki-67-PerCP/Cy5.5 and TLR9-PE. Multiparametric flow cytometric analysis was performed using a Bio-Rad S3e Cell Sorter flow cytometer and analyzed using FlowJo software (Treestar). The population of live cells was detected depending on their size and complexity (FSC-SSC gate) and later the doublets were excluded (FSC-Height/FSC-Area gate).

## Microscopy

For the visualization of tissue sections for RNAscope analysis, the Zeiss Axio Observer.A1 microscope was used. The images were prepared with Photoshop CS6 software.

## Transmission electron microscopy (TEM)

Samples were fixed in 2.5% glutaraldehyde in phosphate buffer (0.1 M, pH = 7.2) for a minimum of 24 hr at 4°C. Then, the samples were washed in phosphate buffer (0.1 M, pH = 7.2), post-fixed in 1% osmium tetroxide, dehydrated in graded ethanol, cleared in propylene oxide, and embedded in an epon/araldite resin mixture and polymerized at 60°C for 24 hr. Semithin sections of 1 μm thickness were cut by using an ultra-microtome (Leica Reichert UCT, Vienna, Austria), stained with toluidine blue, and examined in a light microscope. Silver/gold interference color ultrathin sections were cut and mounted on 300 mesh copper grids, contrasted with uranyl acetate and lead citrate, before being examined under a TEM (TALOS L120C, FEI, USA) operating at 120 kV. Images were collected using the Ceta camera installed on the TEM. Uranyl acetate, Lead Citrate, Toluidine Blue, Araldite Resin, Agar 100 Resin, Dodecenyl Succinic Anhydride (DDSA), Tri-Dimethylaminomethyl Phenol (DMP-30), Osmium tetroxide, Glutaraldehyde, and Copper 300 mesh grids were all purchased from AGAR (Essex, UK).

## Statistical analysis

Nonparametric 2-way ANOVA test was performed and statistical significance was considered at $p < 0.05$. Statistical analyses of the data were performed using the GraphPad Prism v.8.0 (La Jolla, CA). All the experiments were performed using at least three biological replicates.

## Acknowledgements

We thank the members of the Strati (KS) and Kirmizis (AKY) groups for helpful discussions and sharing of materials, reagents and equipments. We also thank Paul F Lambert for the puC19-MmuPV1 plasmid. This work was funded by grants to KS by the Cyprus Research and Innovation Foundation (INFRA-STRUCTURES/1216/0034, OPPORTUNITY/0916/ERC-StG/003, POST-DOC/0916/0111).

## Additional information

### Funding

| Funder | Grant reference number | Author |
| --- | --- | --- |
| Cyprus Research and Innovation Office | INFRASTRUCTURES/1216/0034 | Katerina Strati |
| Cyprus Research and Innovation Office | OPPORTUNITY/0916/ERC-StG/003 | Katerina Strati |

The funders had no role in study design, data collection and interpretation, or the decision to submit the work for publication.

### Author contributions

Vural Yilmaz, Formal analysis, Investigation, Methodology, Validation, Writing – original draft, Writing – review and editing; Panayiota Louca, Investigation, Methodology, Validation, Writing – original draft, Writing – review and editing; Louiza Potamiti, Formal analysis, Formal analysis, Methodology, Writing – review and editing; Mihalis Panayiotidis, Formal analysis, Investigation, Project administration, Writing – review and editing; Katerina Strati, Data curation, Formal analysis, Writing – review and editing, Methodology, Investigation, Writing – original draft, Writing – review and editing

### Author ORCIDs

Vural Yilmaz http://orcid.org/0000-0002-1959-6778
Katerina Strati http://orcid.org/0000-0002-2332-787X

### Ethics

This study was carried out in strict accordance with the recommendations in the Guidelines for the Protection of Laboratory Animals of the Republic of Cyprus. The animal facility is licensed by the

Veterinary Services (Republic of Cyprus Ministry of Agriculture and Natural Resources), the government body in charge of approving and overseeing laboratory animal work in Cyprus (license number CY.EXP.105) and the protocol was approved by the same authority (License number CY/EXP/PR. L1/2019).

## Decision letter and Author response
Decision letter https://doi.org/10.7554/eLife.72638.sa1
Author response https://doi.org/10.7554/eLife.72638.sa2

---

## Additional files

### Supplementary files
• Source data 1. Source data for gels in *Figures 1 and 3*, *Figure 3—figure supplement 1* are provided in a ZIP file called "Source data Gels". Source data for sequences and plasmid creation in detail are provided in a ZIP file called "Source data Sequences and Plasmid creation". The source data of FACS for *Figure 4* and its supplements is included in a Microsoft Excel spreadsheet called "Source data - FACS".

• Source data 2. Source data for sequences and plasmid creation in detail are provided in a ZIP file called "Source data Sequences and Plasmid creation".

• Transparent reporting form

### Data availability
All data generated or analysed during this study are included in the manuscript and supporting file; Source Data files have been provided.

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
