## [Editor Report]

This paper describes the development of an interesting and novel model permitting lineage tracing using the MmuPV1 mouse papillomavirus. This model will allow analysis of the temporal and spatial dynamics of MmuPV1 infection replication and assembly in different tissue sites and is a valuable contribution to the understanding of papillomavirus biology.

---

## [Decision Letter]

**Decision letter after peer review:**

Thank you for submitting your article "A novel lineage-tracing mouse model for MmuPV1 infection enables in vivo studies in the absence of cytopathic effects" for consideration by *eLife*. Your article has been reviewed by 3 peer reviewers, and the evaluation has been overseen by a Reviewing Editor and Diane Harper as the Senior Editor. The following individuals involved in review of your submission have agreed to reveal their identity: Megan E Spurgeon (Reviewer #1); Neil D Christensen (Reviewer #2); Sheila Graham (Reviewer #3).

Essential revisions:

Specifically comments and suggestions from reviewer 1 meet the concerns of all reviewers.

Additional Experiments/Data/Analyses:

1. One of the major pieces of evidence that the MmuPV1-lox-Cre-lox virus can infect murine skin and ultimately establish a productive viral infection is the expression of viral transcripts by RNA in situ hybridization provided in Figure 3. It has been noted by multiple groups that the RNAscope assay using MmuPV1-specific probes can detect MmuPV1 viral DNA. In order to distinguish between viral DNA versus RNA transcripts, treatment with DNAse and/or RNAse can be performed (for example, see Xue et al., 2017 and Spurgeon and Lambert, 2019). The images in Figure 3 are blurry, but it does appear that there is ample nuclear signal in some of the tissues (usually indicative of viral DNA). Because the presence of MmuPV1 viral gene transcription is being used as a readout for active infection and is therefore such a foundational piece of evidence supporting their infection model system, the authors should make an effort to perform DNAse/RNAse treatment in their RNAscope assays to verify the signal is, in fact, due to the presence of MmuPV1 viral transcripts.

2. The production of infectious virus following MmuPV1-lox-Cre-lox delivery and recircularization would provide strong validation of their infection model and act as convincing evidence for its experimental utility. Assays such as L1 protein detection in infected tissues, isolation of viral particles from swabs/tissues, or using swab samples to infect the skin of immunodeficient mice would provide such key data.

3. For the skin swab and subsequent PCR experiment used to detect MmuPV1 L1 and Cre loss shown in Figure 3E, mock-infected control swabs should be included. The gel image quality could also be improved, if possible. In its current form, it is difficult to agree with their conclusion that they detect Cre loss by Day 5.

4. The authors performed flow cytometry to quantify Ki67 positive cells and measured an increase in YFP+ (infected) cells. Direct visualization of Ki67-positive cells by either IF/IHC in the regions of infected skin where they detect RNAscope-positive cells (those shown in Figure 3) would greatly strengthen and complement their data that infected cells exhibit higher proliferation dynamics shown in Figure 4.

Recommendations for improving the writing/presentation:

1. Regarding the title and the words "in the absence of cytopathic effect": This topic is hardly revisited in the Results/Discussion, nor is any rigorous histopathological analysis performed to exclude the presence of cytopathic effect. While their rationale that this will facilitate the study of subclinical infections is legitimate, this phrase does not seem to add anything to the impact of the report or accurately describe the findings discussed therein.

2. Conclusions and Significance (Lines 436-441): Length restrictions being appreciated, discussing a few of the 'outstanding questions in papillomavirus biology' that the authors believe their new infection model can help address would help bolster and convey the importance of their findings.

3. It would be useful to more clearly indicate in the Results and Figure Legends which genetic background is being used in each experiment. It is unclear where C57BL/6 versus FVB mice were used.

4. Lines 69-72: while there are broad similarities between HPVs and MmuPV1 with respect to genome structure, encoded proteins, etc., they are actually quite different at the level of nucleotide sequence (HPV16 vs. MmuPV1 only ~ 50% similar). Likewise, HPV and MmuPV1 E6 and E7 proteins are less than 50% similar at an amino acid level (please refer to Joh et al., J Gen Virol, 2011). While MmuPV1 is certainly a useful tool for studying PV infection and pathogenesis, as is the focus in this report, this text overstates the similarities between HPV and MmuPV1 and should be amended to more accurately reflect the differences.

5. Lines 78-80: It may be worth noting here that subclinical MmuPV1 infections have been observed by several research groups at several anatomical sites. Including this information would strengthen the use of the MmuPV1 infection model to evaluate these understudied types of infection.

6. The images in some of the figures (particularly Figure 2B-C and 3A-B) are blurry and out of focus.

7. The PCR gel image in Figure 3E is very hard to see and is therefore not as convincing as it could be.

8. Line 88: Please provide rationale for infecting tail skin versus other sites of skin (dorsal, ear, etc).

9. Line 136: Please provide technical information regarding the isolation of MEFs or provide a citation.

10. Lines 292-293: The sentence "…only immunocompromised mice are susceptible to the mouse papilloma virus" is not accurate. There are several reports of immunocompetent strains of mice that are susceptible to MmuPV1 infection and disease. It may be more accurate to state that immunocompromised mice are more susceptible.

11. Reference issues:

a. Line 66: Please refer to the original paper that described the discovery of MmuPV1 (Ingle et al., Vet Pathol, 2011) instead of the review article currently cited.

b. Line 77: There is also an inducible mouse model of HPV oncogene expression (see Bottinger et al., Carcinogenesis 2020).

c. Line 77: There are certainly more novel infection models than the Wei et al., 2020 cited here that could be included to accurately reflect the state and scope of MmuPV1 infection models.

*Reviewer #1 (Recommendations for the authors):*

Additional Experiments/Data/Analyses:

1. One of the major pieces of evidence that the MmuPV1-lox-Cre-lox virus can infect murine skin and ultimately establish a productive viral infection is the expression of viral transcripts by RNA in situ hybridization provided in Figure 3. It has been noted by multiple groups that the RNAscope assay using MmuPV1-specific probes can detect MmuPV1 viral DNA. In order to distinguish between viral DNA versus RNA transcripts, treatment with DNAse and/or RNAse can be performed (for example, see Xue et al., 2017 and Spurgeon and Lambert, 2019). The images in Figure 3 are blurry, but it does appear that there is ample nuclear signal in some of the tissues (usually indicative of viral DNA). Because the presence of MmuPV1 viral gene transcription is being used as a readout for active infection and is therefore such a foundational piece of evidence supporting their infection model system, the authors should make an effort to perform DNAse/RNAse treatment in their RNAscope assays to verify the signal is, in fact, due to the presence of MmuPV1 viral transcripts.

2. The production of infectious virus following MmuPV1-lox-Cre-lox delivery and recircularization would provide strong validation of their infection model and act as convincing evidence for its experimental utility. Assays such as L1 protein detection in infected tissues, isolation of viral particles from swabs/tissues, or using swab samples to infect the skin of immunodeficient mice would provide such key data.

3. For the skin swab and subsequent PCR experiment used to detect MmuPV1 L1 and Cre loss shown in Figure 3E, mock-infected control swabs should be included. The gel image quality could also be improved, if possible. In its current form, it is difficult to agree with their conclusion that they detect Cre loss by Day 5.

4. The authors performed flow cytometry to quantify Ki67 positive cells and measured an increase in YFP+ (infected) cells. Direct visualization of Ki67-positive cells by either IF/IHC in the regions of infected skin where they detect RNAscope-positive cells (those shown in Figure 3) would greatly strengthen and complement their data that infected cells exhibit higher proliferation dynamics shown in Figure 4.

Recommendations for improving the writing/presentation:

1. Regarding the title and the words "in the absence of cytopathic effect": This topic is hardly revisited in the Results/Discussion, nor is any rigorous histopathological analysis performed to exclude the presence of cytopathic effect. While their rationale that this will facilitate the study of subclinical infections is legitimate, this phrase does not seem to add anything to the impact of the report or accurately describe the findings discussed therein.

2. Conclusions and Significance (Lines 436-441): Length restrictions being appreciated, discussing a few of the 'outstanding questions in papillomavirus biology' that the authors believe their new infection model can help address would help bolster and convey the importance of their findings.

3. It would be useful to more clearly indicate in the Results and Figure Legends which genetic background is being used in each experiment. It is unclear where C57BL/6 versus FVB mice were used.

4. Lines 69-72: while there are broad similarities between HPVs and MmuPV1 with respect to genome structure, encoded proteins, etc., they are actually quite different at the level of nucleotide sequence (HPV16 vs. MmuPV1 only ~ 50% similar). Likewise, HPV and MmuPV1 E6 and E7 proteins are less than 50% similar at an amino acid level (please refer to Joh et al., J Gen Virol, 2011). While MmuPV1 is certainly a useful tool for studying PV infection and pathogenesis, as is the focus in this report, this text overstates the similarities between HPV and MmuPV1 and should be amended to more accurately reflect the differences.

5. Lines 78-80: It may be worth noting here that subclinical MmuPV1 infections have been observed by several research groups at several anatomical sites. Including this information would strengthen the use of the MmuPV1 infection model to evaluate these understudied types of infection.

6. The images in some of the figures (particularly Figure 2B-C and 3A-B) are blurry and out of focus.

7. The PCR gel image in Figure 3E is very hard to see and is therefore not as convincing as it could be.

8. Line 88: Please provide rationale for infecting tail skin versus other sites of skin (dorsal, ear, etc).

9. Line 136: Please provide technical information regarding the isolation of MEFs or provide a citation.

10. Lines 292-293: The sentence "…only immunocompromised mice are susceptible to the mouse papilloma virus" is not accurate. There are several reports of immunocompetent strains of mice that are susceptible to MmuPV1 infection and disease. It may be more accurate to state that immunocompromised mice are more susceptible.

11. Reference issues:

a. Line 66: Please refer to the original paper that described the discovery of MmuPV1 (Ingle et al., Vet Pathol, 2011) instead of the review article currently cited.

b. Line 77: There is also an inducible mouse model of HPV oncogene expression (see Bottinger et al., Carcinogenesis 2020)

c. Line 77: There are certainly more novel infection models than the Wei et al., 2020 cited here that could be included to accurately reflect the state and scope of MmuPV1 infection models.

*Reviewer #2 (Recommendations for the authors):*

Some items for the authors consideration:

1. The figures include GFP labels for detection of fluorescence – this should be YFP based on expression in the transgenic mice.

2. Evidence for recircularization of MmuPV1 in this model appears to be indirect (viral RNA detection). Future studies could determine whether there is evidence of integration and production of infectious virions (containing circularized viral DNA).

3. Maybe correlate/discuss some of the findings in published RNAseq analyses of various other MmuPV1 (and HPV) infections (both in vitro raft culture studies and tissues).

4. Some typos: l-259 "resulting in YFP expression"; l-84 "MmuPV1"; l-481, Complete reference.

5. L-292 suggests that "only immunocompromised mice are susceptible to the mouse papillomavirus". However, it is clear that all mice are susceptible – it is just that most of the disease in such mice is subclinical and quickly cleared. Perhaps best to indicate that immunocompetent mice are resistant to clinical disease and/or persistence.

6. Interestingly, some of the early work by Campo's group with BPV and HPV identified E5 as a viral protein that down-regulated MHC1 (e.g. Oncogene. 2002 Jan 10;21(2):248-59. doi: 10.1038/sj.onc.1205008. Down-regulation of MHC class I by bovine papillomavirus E5 oncoproteins G Hossein Ashrafi, Emmanouella Tsirimonaki, Barbara Marchetti, Philippa M O'Brien, Gary J Sibbet, Linda Andrew, M Saveria Campo). MmuPV1 is noted to not have an E5 protein, however, other viral proteins impact MHC1 as shown here. Note also that Lamberts group conducted some interesting studies using an E5-expressing transgenic mouse with MmuPV1 infections (Virology. 2020 Feb;541:1-12. doi: 10.1016/j.virol.2019.12.002. Epub 2019 Dec 5. The human papillomavirus 16 E5 gene potentiates MmuPV1-Dependent pathogenesis Alexandra D Torres, Megan E Spurgeon, Andrea Bilger, Simon Blaine-Sauer, Aayushi Uberoi, Darya Buehler, Stephanie M McGregor, Ella Ward-Shaw, Paul F Lambert).

*Reviewer #3 (Recommendations for the authors):*

The introduction is a bit simplistic and could do with much more explanation to clarify for the general readership of the journal. For example, I don't agree that all HPV are commensal since they have an exquisitely tight relationship with the host epithelium and some are pathogenic. In this resepct it would be useful to discuss cutaneous versus mucosa PV infection since we now know of a huge number of commensal γ HPVs. HPVs colonise only epithelia, not "human tissues", which is too general a term. We understand how papillomaviruses enter, replicate and spread in epithelia so this aspect of their biology is understood. We do not know precisely how the life cycle is initiated from a single infected stem cell. So it is important to carefully discuss what is and is not known. Infection of the basal layer of the epithelium via tissue damage may not be necessary for infection: e.g. infection of single layer columnar cells e.g. in the endocervix is likely (see Doorbar and Griffin https://pubmed.ncbi.nlm.nih.gov/30974183/). HPV6/11 laryngeal papillomas and genital warts do show cytopathic effects.

Throughout, the paper is written in quite a superficial manner and lacks proper discussion of relevant literature. There are almost no points of discussion in the paper that are fully explored. Some of the figures/data are hard to see, or to decipher, and could be improved. There is a lack of breadth of data which means that the authors conclusions are not fully substantiated.

For example:

Figure 1 is hard to understand for the Cre non-specialist. A diagram of the plasmids that relate to the predicted band on the gel would be useful and well as annotation of the plasmid diagram in (A) with larger and clearer fonts for the primers. In (D) why is the E4 band the same size and the GAPDH band? Positions of the primers on the MmuPV1 genome should be given in a table. Why is the E6 band so faint compared to the L1 band when transcriptome analyses have shown that E6 is much more highly expressed than L1 (see https://journals.plos.org/plospathogens/article?id=10.1371/journal.ppat.1006715#). Since this is puzzling, these data should be backed up with qRT-PCR data.

UV irradiation of the mice may not affect the innate immune response but as this paper deals with validation of a model data showing the effects of UV on the adaptive and innate immune response in the mice is required.

In Figure 2B there is clear evidence of clonal expansion of MmuPV1 infected cells. However, I don't understand why K14, a basal cell marker is expressed in the upper epithelial layer, apparently in all tissues. Moreover, why is K14 staining colocalising with GFP at Day 3 (which could be explained by de-differentiation of the MmuPV1 cells) but not at subsequent days? It is expected that PV infected cells show decreased expression of differentiation markers such as K10 or involucrin and staining with at least these markers should be carried out to properly characterise the infection.

In Figure 3 the stainings for viral RNAs are quite indistinct. Perhaps removing haematoxylin staining and showing higher magnification would help. H and E staining for each tissue and +/- DNAse in the RNAScope would also be required. The bands for Cre loss and MmuPV1 DNA in Figure 3E are too faint to assess.

The investigation of MHC class I on the cell surface is interesting but the authors do not properly explain the background to this study, namely that HPV E5 is the main mediator of MHC downregulation. HPV E7 had been reported to also contribute. Of course, MmuPV1 does not express E5 but neither (to my knowledge) has MmuPV1 E7 been shown to regulate MHC class 1. The finding that cells surrounding the recombinant MmuPV1-containing cells is indicative, but more work would be necessary to firm up this potentially important conclusion.

---

## [Author Response]

Essential revisions:Specifically comments and suggestions from reviewer 1 meet the concerns of all reviewersAdditional Experiments/Data/Analyses:1. One of the major pieces of evidence that the MmuPV1-lox-Cre-lox virus can infect murine skin and ultimately establish a productive viral infection is the expression of viral transcripts by RNA in situ hybridization provided in Figure 3. It has been noted by multiple groups that the RNAscope assay using MmuPV1-specific probes can detect MmuPV1 viral DNA. In order to distinguish between viral DNA versus RNA transcripts, treatment with DNAse and/or RNAse can be performed (for example, see Xue et al., 2017 and Spurgeon and Lambert, 2019). The images in Figure 3 are blurry, but it does appear that there is ample nuclear signal in some of the tissues (usually indicative of viral DNA). Because the presence of MmuPV1 viral gene transcription is being used as a readout for active infection and is therefore such a foundational piece of evidence supporting their infection model system, the authors should make an effort to perform DNAse/RNAse treatment in their RNAscope assays to verify the signal is, in fact, due to the presence of MmuPV1 viral transcripts.

We have added controls (+DNAse, +DNAse and RNAse) in Figure 3 Supplement 1 panels A and B. These support the initial conclusion that most of the signal detected is a result of RNA transcripts and not due to DNA. We note, that in previous publications (Xue et al., 2017 and Spurgeon and Lambert, 2019) the tissue samples examined were derived from timepoints much later than those examined in our study. It is reasonable to expect that, a few days following genome delivery there is a much lower amount of DNA replication occurring compared to what would be expected in established warts or hyperplastic lesions (Xue et al., 2017)*.*

2. The production of infectious virus following MmuPV1-lox-Cre-lox delivery and recircularization would provide strong validation of their infection model and act as convincing evidence for its experimental utility. Assays such as L1 protein detection in infected tissues, isolation of viral particles from swabs/tissues, or using swab samples to infect the skin of immunodeficient mice would provide such key data.

We agree with the reviewer that determining whether formation of infectious virus is occurring in our system would be interesting. We propose, however, that the main utility of the system is the ability to track the cells which initially received viral genome, the ability to study and compare them with likewise tracked cells in uninfected tissue (systems using in vitro or in vivo generated virions would be superior systems to study infection per se). Nevertheless, we did try to study the formation of virions in our system. Using TEM we were able to visualize virion-like structures in the perinuclear area of epithelial cells (added to Figure 3F). None of those structures were observed in mice treated with the control CAG-Cre plasmid. We provide evidence for capsid formation (Figure 3F), genome recircularization (Figure 3E), early and late gene transcription (Figure 3A-B). However, presumably due to the low amounts of virus replication in the time points studied, and the genetic background of the mice infected which we examined here (C57BL/6), we were unable to detect L1 using a commercially available antibody (Papillomavirus antiserum polyclonal goat antibody, ViroStat, 5001, 1:100), and we were not able to infect immunocompromised animals from swabs of infected mice.

3. For the skin swab and subsequent PCR experiment used to detect MmuPV1 L1 and Cre loss shown in Figure 3E, mock-infected control swabs should be included. The gel image quality could also be improved, if possible. In its current form, it is difficult to agree with their conclusion that they detect Cre loss by Day 5.

We have added controls from the mock infected control swabs (Figure 3 Supplement 1C). We continue to detect very low amounts of recombined plasmid which we believe reflects the limited amount of recombined genome present 5 days post-infection. Detection of recombined plasmid is a lot more robust in Figure 1B where we were able to collect cells in culture instead of swabs of superficial epithelia.

4. The authors performed flow cytometry to quantify Ki67 positive cells and measured an increase in YFP+ (infected) cells. Direct visualization of Ki67-positive cells by either IF/IHC in the regions of infected skin where they detect RNAscope-positive cells (those shown in Figure 3) would greatly strengthen and complement their data that infected cells exhibit higher proliferation dynamics shown in Figure 4.

We provide IHC for Ki67 (Figure 4D) which supports conclusions made using FACS on day 5 of infection (more positive staining in basal and suprabasal cells).

Recommendations for improving the writing/presentation:1. Regarding the title and the words "in the absence of cytopathic effect": This topic is hardly revisited in the Results/Discussion, nor is any rigorous histopathological analysis performed to exclude the presence of cytopathic effect. While their rationale that this will facilitate the study of subclinical infections is legitimate, this phrase does not seem to add anything to the impact of the report or accurately describe the findings discussed therein.

We have removed the words “in the absence of cytopathic effect” from the title.

2. Conclusions and Significance (Lines 436-441): Length restrictions being appreciated, discussing a few of the 'outstanding questions in papillomavirus biology' that the authors believe their new infection model can help address would help bolster and convey the importance of their findings.

We have added the relevant discussion (lines 267-270). Briefly, we believe that the novel model described here can be used to investigate the early effects of papillomavirus on the target cells, transcriptional and phenotypic changes of the papillomavirus-harboring cells during the acute phase of the infection, and investigate the initial viral-host interactions.

3. It would be useful to more clearly indicate in the Results and Figure Legends which genetic background is being used in each experiment. It is unclear where C57BL/6 versus FVB mice were used.

We have made the suggested clarifications concerning the genetic background of the mice used in the experiments. All in vivo experiments were performed with R26R-lox-STOP-lox-eYFP immunocompetent mice of pure C57BL/6 genetic background. This is indicated in the methods section (line 340-343). No FVB mice were used in this manuscript.

4. Lines 69-72: while there are broad similarities between HPVs and MmuPV1 with respect to genome structure, encoded proteins, etc., they are actually quite different at the level of nucleotide sequence (HPV16 vs. MmuPV1 only ~ 50% similar). Likewise, HPV and MmuPV1 E6 and E7 proteins are less than 50% similar at an amino acid level (please refer to Joh et al., J Gen Virol, 2011). While MmuPV1 is certainly a useful tool for studying PV infection and pathogenesis, as is the focus in this report, this text overstates the similarities between HPV and MmuPV1 and should be amended to more accurately reflect the differences.

We amended the particular statement to more accurately represent the differences as well as similarities between HPVs and MmuPV1 (Lines 69-74). The relevant citation has been included (Joh et al., J Gen Virol, 2011).

5. Lines 78-80: It may be worth noting here that subclinical MmuPV1 infections have been observed by several research groups at several anatomical sites. Including this information would strengthen the use of the MmuPV1 infection model to evaluate these understudied types of infection.

We have mentioned and appropriately cited the previous reporting of subclinical MmuPV1 infections by other groups (lines 83-84).

6. The images in some of the figures (particularly Figure 2B-C and 3A-B) are blurry and out of focus.

We made several changes to the images presented in the Figures 2 and 3, such as enhancing or replacing the blurry and out of focus ones.

7. The PCR gel image in Figure 3E is very hard to see and is therefore not as convincing as it could be.

We made effort to increase the overall quality of the gels presented here. However, as we mentioned earlier, we detect very low amounts of recombined plasmid which we believe reflects the limited amount of genome hence the faint bands we observe at these time points. Detection of recombined plasmid is a lot more robust in Figure1B where we were able to collect cells in culture instead of swabs of superficial epithelia.

8. Line 88: Please provide rationale for infecting tail skin versus other sites of skin (dorsal, ear, etc).

Tail skin was selected for practical reasons (ability to track area where plasmid was delivered, fewer hairs which place barriers on cell dissociation for FACS experiments). This rationale is added to the Materials and methods section (lines 361-364).

9. Line 136: Please provide technical information regarding the isolation of MEFs or provide a citation.

We have provided technical information regarding the isolation, handling and usage of MEFs (Lines 307-323).

10. Lines 292-293: The sentence "…only immunocompromised mice are susceptible to the mouse papilloma virus" is not accurate. There are several reports of immunocompetent strains of mice that are susceptible to MmuPV1 infection and disease. It may be more accurate to state that immunocompromised mice are more susceptible.

We made the following change to the mentioned sentence “…immunocompromised mice are more susceptible to the mouse papilloma virus.” Lines 138-139.

11. Reference issues:a. Line 66: Please refer to the original paper that described the discovery of MmuPV1 (Ingle et al., Vet Pathol, 2011) instead of the review article currently cited.b. Line 77: There is also an inducible mouse model of HPV oncogene expression (see Bottinger et al., Carcinogenesis 2020).c. Line 77: There are certainly more novel infection models than the Wei et al., 2020 cited here that could be included to accurately reflect the state and scope of MmuPV1 infection models.

Suggested corrections to references have been made (line 66, line 79, line 79-81 respectively).

Reviewer #3 (Recommendations for the authors):The introduction is a bit simplistic and could do with much more explanation to clarify for the general readership of the journal. For example, I don't agree that all HPV are commensal since they have an exquisitely tight relationship with the host epithelium and some are pathogenic. In this resepct it would be useful to discuss cutaneous versus mucosa PV infection since we now know of a huge number of commensal γ HPVs. HPVs colonise only epithelia, not "human tissues", which is too general a term. We understand how papillomaviruses enter, replicate and spread in epithelia so this aspect of their biology is understood. We do not know precisely how the life cycle is initiated from a single infected stem cell. So it is important to carefully discuss what is and is not known. Infection of the basal layer of the epithelium via tissue damage may not be necessary for infection: e.g. infection of single layer columnar cells e.g. in the endocervix is likely (see Doorbar and Griffin https://pubmed.ncbi.nlm.nih.gov/30974183/). HPV6/11 laryngeal papillomas and genital warts do show cytopathic effects.

We have made changes to the introduction as per the reviewers’ suggestions (e.g. “human tissues” was changed to “human epithelia”). We agree with this reviewer that not all HPVs are commensal. Our discussion focuses on the non-pathogenic infections being an understudied area of PV biology and make explicit mentions to study of PV-induced carcinogenesis (for example: line 46-47 “cause 5% of human cancers and are also responsible for commensal infections”..). We have ensured that our introduction does not appear to misrepresent all papillomavirus infections as commensal.

Unfortunately, the word limit prohibits us from extensively expanding the introduction or any other section of this paper.

Throughout, the paper is written in quite a superficial manner and lacks proper discussion of relevant literature. There are almost no points of discussion in the paper that are fully explored. Some of the figures/data are hard to see, or to decipher, and could be improved. There is a lack of breadth of data which means that the authors conclusions are not fully substantiated.

This manuscript was submitted as a short report (“high impact, small size”). We have made every effort to improve the points raised by the reviewers within length constraints. The current length in terms of words and figures is on the upper limit suggested by the journal.

For example:Figure 1 is hard to understand for the Cre non-specialist. A diagram of the plasmids that relate to the predicted band on the gel would be useful and well as annotation of the plasmid diagram in (A) with larger and clearer fonts for the primers. In (D) why is the E4 band the same size and the GAPDH band? Positions of the primers on the MmuPV1 genome should be given in a table. Why is the E6 band so faint compared to the L1 band when transcriptome analyses have shown that E6 is much more highly expressed than L1 (see https://journals.plos.org/plospathogens/article?id=10.1371/journal.ppat.1006715#). Since this is puzzling, these data should be backed up with qRT-PCR data.

As suggested by the reviewer we have added an annotated plasmid diagram to better explain primer location and expected PCR product sizes. Regarding questions about Figure 1D (E6 vs L1 band) these bands derive from separate PCR reactions in cultured cells and are not directly comparable to transcript levels from a mouse infection model or to each other.

UV irradiation of the mice may not affect the innate immune response but as this paper deals with validation of a model data showing the effects of UV on the adaptive and innate immune response in the mice is required.

The role of UV irradiation has been previously described (Uberoi et al., 2016). We believe that further study into the effects of UV in the innate and adaptive response of the mouse, while interesting, would be outside the scope of the current study.

In Figure 2B there is clear evidence of clonal expansion of MmuPV1 infected cells. However, I don't understand why K14, a basal cell marker is expressed in the upper epithelial layer, apparently in all tissues. Moreover, why is K14 staining colocalising with GFP at Day 3 (which could be explained by de-differentiation of the MmuPV1 cells) but not at subsequent days? It is expected that PV infected cells show decreased expression of differentiation markers such as K10 or involucrin and staining with at least these markers should be carried out to properly characterise the infection.

In order to preserve epitopes for YFP staining we have used frozen sections instead of paraffin embedded ones. In our experience, frozen sections are not optimal for use of antibodies to K14 etc. K14 staining is only useful to provide orientation to epithelium but not make conclusive characterization of tissue differentiation status in this context. We kindly remind this reviewer that this paper was submitted as a short report and is at the upper limit in terms of length.

In Figure 3 the stainings for viral RNAs are quite indistinct. Perhaps removing haematoxylin staining and showing higher magnification would help. H and E staining for each tissue and +/- DNAse in the RNAScope would also be required. The bands for Cre loss and MmuPV1 DNA in Figure 3E are too faint to assess.

As mentioned above, we have added controls (+DNAse, +DNAse and RNAse) in Figure 3 Supplement 1 panels A and B. These support the initial conclusion that most of the signal detected is a result of RNA transcripts and not due to DNA. Additionally, we made several changes to the images presented in the Figures 2 and 3, such as enhancing or replacing the blurry and out of focus ones. As for the bands for Cre loss and MmuPV1 DNA in Figure 3E, we continue to detect very low amounts of recombined plasmid which we believe reflects the limited amount of genome present 5 days post-infection.

The investigation of MHC class I on the cell surface is interesting but the authors do not properly explain the background to this study, namely that HPV E5 is the main mediator of MHC downregulation. HPV E7 had been reported to also contribute. Of course, MmuPV1 does not express E5 but neither (to my knowledge) has MmuPV1 E7 been shown to regulate MHC class 1. The finding that cells surrounding the recombinant MmuPV1-containing cells is indicative, but more work would be necessary to firm up this potentially important conclusion.

We agree with the reviewer that our interesting finding of decreased MHC-I expression is certainly in need of further investigation to discover the underlying mechanims. However, given the nature of a short report paper, our purpose here is to provide an example of usage of the described novel infection model which can be used to answer such questions about the interplay between virus and host immune system. Our future work will certainly include such investigations by using this novel model and determine the possible role of MmuPV1 -E7 in regulating immune responses to MmuPV1.